# *C. elegans* genome-wide analysis reveals DNA repair pathways that act cooperatively to preserve genome integrity upon ionizing radiation

Bettina Meier[1☉], Nadezda V. Volkova[2☉], Bin Wang[1,3], Víctor González-Huici[1¤], Simone Bertolini[1], Peter J. Campbell[4,5,6], Moritz Gerstung[2,7]*, Anton Gartner[1,8,9]*

**1** Centre for Gene Regulation and Expression, University of Dundee, Dundee, United Kingdom, **2** European Molecular Biology Laboratory, European Bioinformatics Institute, Hinxton, United Kingdom, **3** National Engineering Research Center for Non-Food Biorefinery, Guangxi Academy of Sciences, Nanning, China, **4** Cancer, Ageing and Somatic Mutation, Wellcome Sanger Institute, Hinxton, United Kingdom, **5** Department of Haematology, University of Cambridge, Cambridge, United Kingdom, **6** Department of Haematology, Addenbrooke's Hospital, Cambridge, United Kingdom, **7** European Molecular Biology Laboratory, Genome Biology Unit, Heidelberg, Germany, **8** Center for Genomic Integrity, Institute for Basic Science, Ulsan, Republic of Korea, **9** Department of Biological Sciences, School of Life Sciences, Ulsan National Institute of Science and Technology, Ulsan, Republic of Korea

☉ These authors contributed equally to this work.
¤ Current address: Institute for Research in Biomedicine, Parc Científic de Barcelona, Barcelona, Spain
* tgartner@ibs.re.kr (AG); moritz.gerstung@ebi.ac.uk (MG)

**Data Availability Statement:** Sequencing data are available under ENA Study Accession Numbers ERP000975 and ERP004086 with ENA sample IDs

## Abstract

Ionizing radiation (IR) is widely used in cancer therapy and accidental or environmental exposure is a major concern. However, little is known about the genome-wide effects IR exerts on germ cells and the relative contribution of DNA repair pathways for mending IR-induced lesions. Here, using *C. elegans* as a model system and using primary sequencing data from our recent high-level overview of the mutagenic consequences of 11 genotoxic agents, we investigate in detail the genome-wide mutagenic consequences of exposing wild-type and 43 DNA repair and damage response defective *C. elegans* strains to a Caesium (Cs-137) source, emitting γ-rays. Cs-137 radiation induced single nucleotide variants (SNVs) at a rate of ~1 base substitution per 3 Gy, affecting all nucleotides equally. In nucleotide excision repair mutants, this frequency increased 2-fold concurrently with increased dinucleotide substitutions. As observed for DNA damage induced by bulky DNA adducts, small deletions were increased in translesion polymerase mutants, while base changes decreased. Structural variants (SVs) were augmented with dose, but did not arise with significantly higher frequency in any DNA repair mutants tested. Moreover, 6% of all mutations occurred in clusters, but clustering was not significantly altered in any DNA repair mutant background. Our data is relevant for better understanding how DNA repair pathways modulate IR-induced lesions.

annotated in Supplementary S1 Table. The R code for analysis of mutation rates, mutation signatures, and mutation clustering is available on GitHub under http://github.com/gerstung-lab/radiation.

**Funding:** This work was supported by the Wellcome Trust [COMSIG consortium grant RG70175 to A.G., Senior Research award 090944/Z/09/Z to A.G.]; Worldwide Cancer Research [18-0644 to A.G.]; and the Korean Institute for Basic Science [IBS-R022-A2-2021 to A.G.]. Nadezda Volkova is a member of Lucy Cavendish College, University of Cambridge. The funders had no role in study design, data collection and analysis, decision to publish, or preparation of the manuscript.

**Competing interests:** The authors have declared that no competing interests exist.

## Introduction

Genome integrity is essential for cellular and organismal survival. One of the most important genotoxic agents is IR. Irradiation occurs during environmental or accidental exposure, during diagnostic radiology or cancer radiotherapy. Exposure to radon, a naturally occurring radioactive noble gas produced by the decay of Uranium-226 in the earth's crust, is reported to be associated with ~10% of lung cancers [1]. Health consequences of IR exposure were studied on survivors of atomic bomb explosions [2–4] and the Chernobyl nuclear reactor accident that led to the release of large quantities of Iodine-131 and Caesium-137 [5]. Long-term effects of radiation exposure are evaluated epidemiologically as increased cancer incidence, an undertaking hampered by latency periods that can span several decades and the difficulty to establish causal relationships when the exposure to radiation is low. Utilizing the genotoxic potential of radiation, radiotherapy is a highly effective cancer treatment, first employed within a year after the discovery of IR in the late 19[th] century, and currently applied to treat ~40% of all UK cancer patients [6].

The effects of IR on inheritance were first determined by Herman Müller in the 1920s, recording increased frequencies of lethal and visible mutations in fruit flies [7]. In *C. elegans*, counting the number of lethal mutations associated with 'chromosomal translocations' allowed for the frequency of such events to be gauged at 0.12 per gamete per 15 Gy of IR [8, 9]. More recently, genome-wide sequence analyses of IR exposure have started to emerge. In budding yeast, whole genome sequencing detected ~2 single nucleotide substitutions following exposure to 100 Gy of γ-rays [10]. The use of IR to induce mutations for plant breeding motivated studies in several plant species including *Arabidopsis*, *rice* and *banana*. In *Arabidopsis*, 200 Gy of carbon ion beam radiation yielded 20–60 single nucleotide variants (SNVs) [11], and 60 Gy of fast neutron radiation led to a range of 8–32 mutations in 6 sequenced lines [12]. In rice, sequencing of 41 lines derived from irradiated parental plants treated with 20 Gy of fast neutrons yielded an average of ~31 SNVs, ~3.5 deletions, ~3.5 insertions, ~2 inversions and ~4.2 translocations per line [13]. In banana plants, large deletions were observed in ~⅔ of lines subjected to 20 Gy or 40 Gy of γ-radiation from a cobalt-60 source [14]. A comparison of the mutation rates in the progeny of mice exposed to 3 Gy of X-rays and unirradiated controls did not identify elevated numbers of SNVs, but a ~2 fold increased frequency of indels and an average of 0.08 copy number variants (CNVs) [15]. These results are broadly in line with a recent study confirming the absence of IR-induced SNVs, but reporting on average 9.6 radiation-induced indels in the progeny derived from mice spermatogonia irradiated with 4 Gy [16]. In contrast, the analysis of 12 human secondary malignancies associated with radiation treatment revealed an average of ~4000 SNVs with all four DNA bases mutated at equal frequency [17]. In addition, ~200 deletions smaller than 100 bp in size, often with microhomology at their breakpoints, were observed [17]. The radiation doses applied to treat the corresponding primary malignancies were not reported, but cumulative cancer radiotherapy doses typically range from 20 to 80 Gy [18]. More recent studies, analysing thyroid carcinomas, linked to Iodine-131 exposure related to the Chernobyl disaster failed to detect increased SNV levels, indels and SVs being increased [19]. Human organoid exposure to IR similarly only led to increased indels and SVs [20]. In radiation induced tumours induced in p53 mutant mice SNVs levels were reported to be rare, indels and SV being detectable [21].

In summary, while these analyses provide insights into mutation rates and mutation types, their comparison is complicated by the types of analysis applied, differences in effects of chosen radiation sources, DNA repair status, tissue type, and possible organismal differences. Also, these studies do not provide insight into the relative contribution of various DNA repair pathways counteracting IR induced mutagenesis.

IR directly damages DNA by causing breaks in the DNA backbone or hydrogen bonds between bases, or by damaging bases. In addition, approximately two thirds of all damage caused by X- and γ-rays is thought to be caused by reactive oxygen species that form when water or organic molecules are hit [22, 23]. In human cells, 1 Gy of radiation is thought to induce 2000 base damages, 1000 DNA single-strand breaks, 40 DNA double-strand breaks, and 150 DNA-protein crosslinks [24, 25]. Modelling of radiation track structures with X- or γ-rays, led to two or more ionizations within 1–4 nm a distance that corresponds to the diameter of the DNA double helix and associated water layers, thus potentially inducing clustered DNA lesions [23–25]. The high quantity and multifarious nature of DNA damage inflicted by IR, in contrast to the relatively low mutation rates reported, imply that cells utilize their arsenal of DNA damage repair pathways to efficiently restore their genetic information from the vast majority of IR-induced lesions. To the best of our knowledge, the contributions of various DNA repair and damage response pathways to mending DNA damage caused by IR has not been systematically investigated on a genome-wide scale.

We recently reported a global, high-level analysis encompassing 2700 *C. elegans* genomes treated with 11 genotoxic agents [26]. This study demonstrated that DNA repair activities play a major role in shaping mutagenesis. Altered mutation rates and spectra were observed in 40% of analysed DNA repair mutants from various DNA repair pathways [26]. Here, using the same primary sequence data, we expand the analysis to provide a detailed and comprehensive investigation of mutagenesis induced by exposure to Cs-137. We dissect the role of DNA repair and DNA damage response factors in preventing or modulating IR-induced mutagenesis through mutation rate calculations and mutation profile analysis. Our study provides a comprehensive catalogue of radiation damage in wild-type and DNA repair deficient *C. elegans*. Our study highlights that the number of mutations (especially SVs) inflicted by IR is surprisingly low and that determining the contribution of various DNA repair pathways in preventing SVs is challenging. Our data are consistent with the notion that the vast majority of IR-induced lesions are repaired by several, redundant DNA repair pathways. Our results are relevant for a better understanding of the genome-wide effects of accidental or intended IR exposure and may influence clinical strategies to sensitize tumour cells to IR-induced cell death by targeting the most protective DNA repair pathways.

## Materials and methods

### *C. elegans* strains

*C. elegans* mutants used in this study (S1 Table) were backcrossed 6 times against the wild-type N2 reference strain TG1813 [27] and frozen as glycerol stocks.

### Treatment of *C. elegans* strains and sample preparation for sequencing

L4 stage animals were irradiated with doses of 20 Gy, 40 Gy, 60 Gy and 80 Gy using a Cs-137 source (IBL 437C, CIS Bio International), and doses of 50 J, 100 J, 200 J and 500 J using a Waldman UV 236 B device with a spectrum of 280 nm to 360 nm (UV-B / UV-A light). Cisplatin exposure was performed as described [27]. In short, animals were incubated under gentle shaking for 16h at 20°C in M9 liquid culture with cisplatin (Sigma-Aldrich, P4394) concentrations from 0 to 400 μM (S1 Table). Treated animals were allowed to recover and 3 times 3 adults per genotype and dose were transferred onto fresh plates 24h post treatment. Adults were removed after 4h and eggs laid within this time were allowed to hatch and scored for viability [26, 27]. 2 F1 L4s per plate were then singled. For higher doses of radiation, not all F1 animals produced progeny. Progeny of only one of each set of 2 F1 lines was frozen, providing 3 independent lines per genotype and treatment. Genomic DNA was isolated using

Invitrogen ChargeSwitch® gDNA Mini Tissue Kit (Thermo Fisher Scientific, CS11204) and sent for sequencing.

## Variant analysis and mutation rate estimation

DNA samples were sequenced using Illumina HiSeq 2000 100 bp short read paired-end sequencing. Alignment, variant calling, post-processing filtering, and data analysis were performed, as described previously [26, 28]: raw reads were aligned against WBcel235.74.dna. toplevel.fa reference genome (http://ftp.ensembl.org/pub/release-74/fasta/caenorhabditis_ elegans/dna/) using BWA [29], and variant analysis was performed using CaVEMan, Pindel and DELLY [30–32]. A parental wild-type line was used as a control for variant calling. Additional filtering included applying cut-offs on coverage, read support of the variant in the test and control samples, and overlap with other genomic variants (particularly applicable for base changes and indels in homopolymer runs), as previously described [26]. Variants of each sample were additionally filtered against a panel of unrelated samples to exclude technical artifacts (for more details, see S4 File). Detailed information of samples used in this study and their corresponding ENA accession codes are provided in S1 Table.

## Statistical analysis of mutation rates and signatures

Average numbers of SNVs, MNVs, indels, SVs and overall numbers of mutations per dose were calculated using non-zero intercept additive Poisson regression across all samples and radiation doses for a given genotype. Effects of different genetic backgrounds, the mutational signature of IR, as well as the mutation rate fold-changes per genotype were inferred following a model almost identical to the one described in [26]: we modelled observed mutation counts in 119 mutation types (96 SNVs based on their trinucleotide contexts, 14 indels of different types and sizes, and 7 types of SVs) using additive Poisson regression with separate components for genomic contribution and IR: for a genetic background G, $Y_G \sim$ Poisson($G$ + dose·$IR_{wt}$·exp($LFC_G$), where G is a 119-long vector denoting the background contribution shared across all samples of the same genotype, $IR_{wt}$ denotes mutational signature of IR in wild-type, and $LFC_G \in \mathbb{R}^{119}$ denotes log fold-change between IR signature in background G and in wild-type. Estimates for the mean and standard deviation of the parameters were obtained from HMC sampling of the observed counts.

Significance of the differences between mutation rates in different mutation classes was assessed based on z-test comparing the coefficients estimated from additive Poisson model for each DNA repair deficient background and that for wild-type [33]. Statistical comparisons across genotypes were done applying false discovery rate (FDR) control using the Benjamini-Hochberg procedure to correct for multiple testing [34].

## Analysis of clustered mutations

Clustering of mutations was assessed using the start points of all base substitutions and indels across samples of the same genotype and generation. Clustered status was assigned based on a sliding window of 1000 bps.

To obtain the rates of clustering per dose, we used a linear model: $\mathbf{P} \sim \mathbf{proportions} + \varepsilon, \varepsilon \sim$ N(0,$\sigma^2$). Clustering estimates in DNA repair deficient backgrounds were compared to those in wild-type by the following Z test:

$Z = \frac{r_g - r_{wt}}{\sqrt{SE(r_g)^2 + SE(r_{wt})^2}}$ [33]. False discovery rate among the resulting p-values was corrected for multiple testing using Benjamini-Hochberg procedure [34].

## Cancer data analysis

Analysis of cancer samples associated with radiation exposure was performed using the WGS data for 12 radiation-associated secondary malignancies from [17]. Data for bone and breast cancers without an obvious association to IR exposure was obtained from the ICGC data portal (https://icgc.org) by selecting WGS-analysed primary tumour samples with less than 10,000 mutations/genome from BRCA-UK (33 samples) and BOCA-UK (62 samples) projects. Mutation cluster analysis was performed following the same procedure as described above using different transition probabilities: 0.01 for transition from clustered to non-clustered state and 0.1 for an inverse transition. For mutational spectra comparison, *C. elegans* mutation counts were adjusted to the differences in human trinucleotide composition [28].

## Results

### IR-induced mutations in *C. elegans* wild-type

Mutation spectra derived from wild-type *C. elegans* exposed to increasing doses of IR in independently conducted, triplicate experiments, revealed a linear and dose-dependent increase of single nucleotide variants (SNVs) (Figs 1B and 2A and S1 Table, list of *C. elegans* strains and samples). Indels and structural variants (SVs) were also increased, albeit to a lesser extent, occurring with a lower incidence (Figs 1B and 2A). Using additive Poisson regression on mutations across all wild-type samples, we calculated an average number of 36.63 (SD = 1.13) SNVs, 1.3 (SD = 0.2) DNVs, and 4 (SD = 0.4) indels and 1.4 (SD = 0.13) SVs per 80 Gy (Fig 1B) (Materials and Methods). Mutation distribution was uniform across the 6 possible SNVs C>A, C>G, C>T, T>A, T>C and T>G (Figs 1B and 3B, top panel) with no overt influence of adjacent 5' or 3' nucleotides [26]. A similarly flat SNV pattern has previously been reported in radiation-associated human cancers [17]. To enable comparison between our results and human cancer profiles, we adjusted the *C. elegans* mutational SNV pattern to trinucleotide frequencies in the human genome, as described previously (Fig 1C) [28]. Using the humanized *C. elegans* pattern, we identified three human cancer mutation profiles with high similarity; the SNV pattern observed in radiation-associated secondary tumours (cosine similarity = 0.85) [17], COSMIC signature 3 (cosine similarity = 0.84), and COSMIC signature 40 (cosine similarity = 0.92). COSMIC signatures were computationally extracted from thousands of cancer genomes encompassing most cancer types [35]. COSMIC signature 3 has been associated with HR deficiency, while signature 40 is increased with age and is detectable in a large variety of tumours where it typically only contributes a small proportion of mutations. The reduced frequency of C>A, C>G, C>T changes in the context of G as the 3' nucleotide visible in the humanised *C. elegans* patterns and three related cancer signatures (Fig 1C) can be explained by the reduced abundance of CG sequences in the mammalian genome resulting from 5-methylcytosine deamination. Interestingly, mutational signatures derived from HR defective *C. elegans* lines or DT40 cells propagated over multiple generations without exposure to DNA damaging agents, also show a uniform SNV profile in conjunction with small deletions and SVs [28, 36]. Among deletions, 1 bp deletions were most common, followed by 2–5 and 6–50 bp deletions, and >50 bp deletions in *C. elegans* and radiation-associated secondary tumours (S1A Fig) [17]. Turning to SVs, we detected 0–2 SVs in irradiated wild-type lines (Figs 1B and S1B). The proportion of SV types varied between irradiated nematodes and radiation-associated secondary tumours; deletions occurred more frequently in *C. elegans* whereas inversions and translocations were more prominent in human cancer (S1B Fig). In summary, the pattern of IR-induced SNVs is evenly spread across all base changes in *C. elegans* wild-type and no overt influence of 5' and 3' bases could be observed. In addition, IR-exposure induced indels

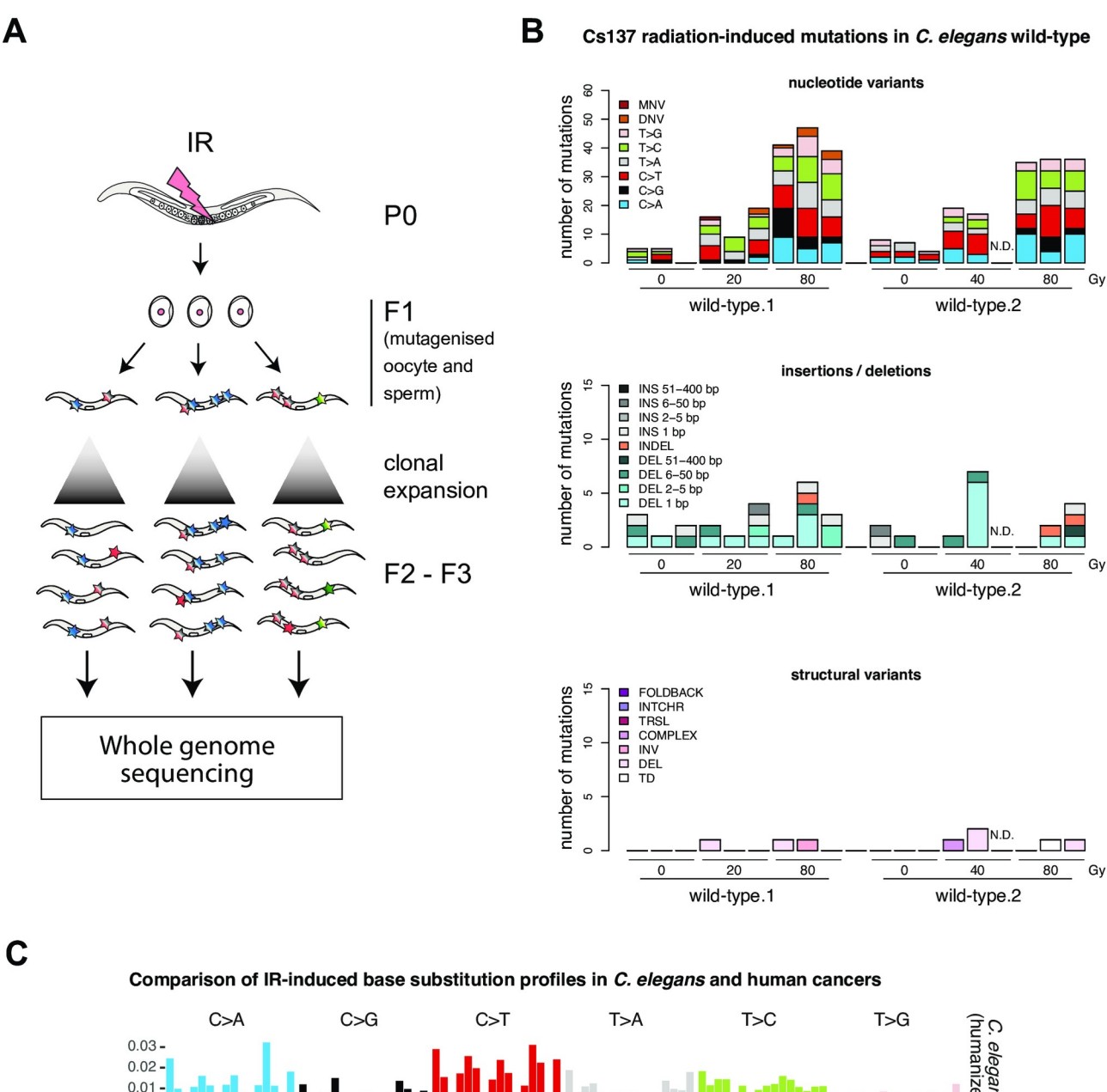

**Fig 1. Experimental outline and mutation patterns induced by ionising radiation in *C. elegans* wild-type compared to human cancer samples. A.** Experimental outline: Parental (P0) animals were exposed to a Cs137 radiation source at the L4 developmental stage. 3 times 3 P0 animals were allowed to lay eggs and two L4s of the first filial (F1) generation per P0 plate were individually transferred to new plates and allowed to clonally expand by self-

fertilization. One F1 line per P0 plate was used for DNA preparation and whole genome sequencing (Materials and Methods). **B.** The number of mutations observed in two biological replicates exposed to increasing doses of Cs-137 radiation are shown. Mutations are categorized into base substitutions (upper panel) as 6 possible single nucleotide variants (SNV), dinucleotide variants (DNV) and multi-nucleotide variants (MNVs), indels (center panel) as insertions (INS), deletions (DEL) and deletions with insertions (INDEL), and structural variants (SVs) (lower panel) encompassing tandem duplications (TD), large deletions (DEL), inversions (INV), complex (COMPLEX) SVs, translocations (TRSL), interchromosomal rearrangements (INTCHR) and foldback structures (FOLDBACK). **C.** Comparison of the IR-induced *C. elegans* SNV profile adjusted to the human trinucleotide content with mutation profiles observed in radiation-exposed human secondary tumours (human) and COSMIC signatures 3 (Sig 3) and 40 (Sig 40) [17, 35]. SNV profiles are depicted as the relative contribution of the six possible SNVs in their 5' and 3' base context. Cosine similarities between the humanized *C. elegans* signature and human cancer profiles are shown.

and SVs. The comparison between the γ-ray treated *C. elegans* and x-ray treated human cancer samples may thus suggest that both IR sources induce base damage and subsequent substitutions that are largely independent of sequence context.

## Contribution of DNA repair deficiencies to IR-induced mutation patterns

To systematically explore how various DNA repair pathways contribute to the repair of Cs-137-inflicted DNA damage, we investigated mutation rates and spectra from 48 DNA repair and DNA damage response deficient *C. elegans* strains (S1 Table) [26]. These include strains defective for apoptosis, p53, base excision repair (BER), single-strand break repair (SSBR), nucleotide excision repair (NER), mismatch repair (MMR), direct damage reversal mediated by O6-methyl guanine methyl transferases (MGMTs), double-strand break repair (DSBR) by homologous recombination (DSBR-HR), non-homologous end-joining (DSBR-NHEJ) and microhomology-mediated DNA end-joining (DSBR-MMEJ), DNA crosslink repair (CL), and mutants defective for translesion synthesis (TLS) polymerases [26] (S1 Table). Overall, all DNA repair mutants proliferated normally under unchallenged conditions. Upon irradiation, we observed a high level of variation when comparing mutation numbers between individual irradiated lines, especially for indels and structural variants (Fig 2A, wild-type; S1 File, all genotypes). We thus applied rigorous criteria to assess which genotypes led to significantly increased or decreased rates of SNVs, indels and SVs in response to IR, requiring that at least two classes of mutations show a dose dependent increase (Fig 2A, wild-type; S1 File, all geno-types). 35/44 mutants fulfilled these dose response criteria and were subsequently analysed in detail (S2–S5 Figs, for raw mutation count data for all 44 strains by DNA repair and damage response pathway). We found that mutation numbers and mutational patterns induced by Cs-137 did not vary greatly across most genotypes. Of the 35 mutants analysed, 10 displayed significantly elevated mutagenesis (at 5% FDR) (Figs 2B and 3, for significantly different mutation profiles, S2 File, for all mutation profiles). Strains with mutational patterns significantly different from wild-type are indicated by an asterisk (*) ($P \leq 0.05$), strains excluded from our detailed analysis are indicated by ∉.

## DNA repair defects associated with increased levels of SNVs

SNV numbers and indels were significantly elevated in *xpa-1*, *xpc-1*, and *xpf-1* NER mutants (Figs 2B and S2A). XPF-1 and XPA-1 contribute to both global genome (GG-NER) and transcription coupled NER (TC-NER), XPC-1 is solely involved in GG-NER, and CSB-1 specifically contributes to TC-NER. Our finding that mutagenesis was increased in *xpa-1*, *xpc-1*, and *xpf-1* but not in *csb-1* mutants (Figs 2B and S2 and S2 File) indicates that a high proportion of IR-induced base changes and indels were prevented by error-free GG-NER.

SNV numbers and indels were also increased in *brc-1* HR defective lines (Figs 2B and 3B and S3). Indels (but not base substitutions) were also increased in *rad-54*, *slx-1* and *mrt-2* HR defective mutants (Figs 2B and 3B and S2 File and S3 Fig) and a strain defective for the *ung-1*

**A** Dose-dependent mutation burden in *C. elegans* wild-type

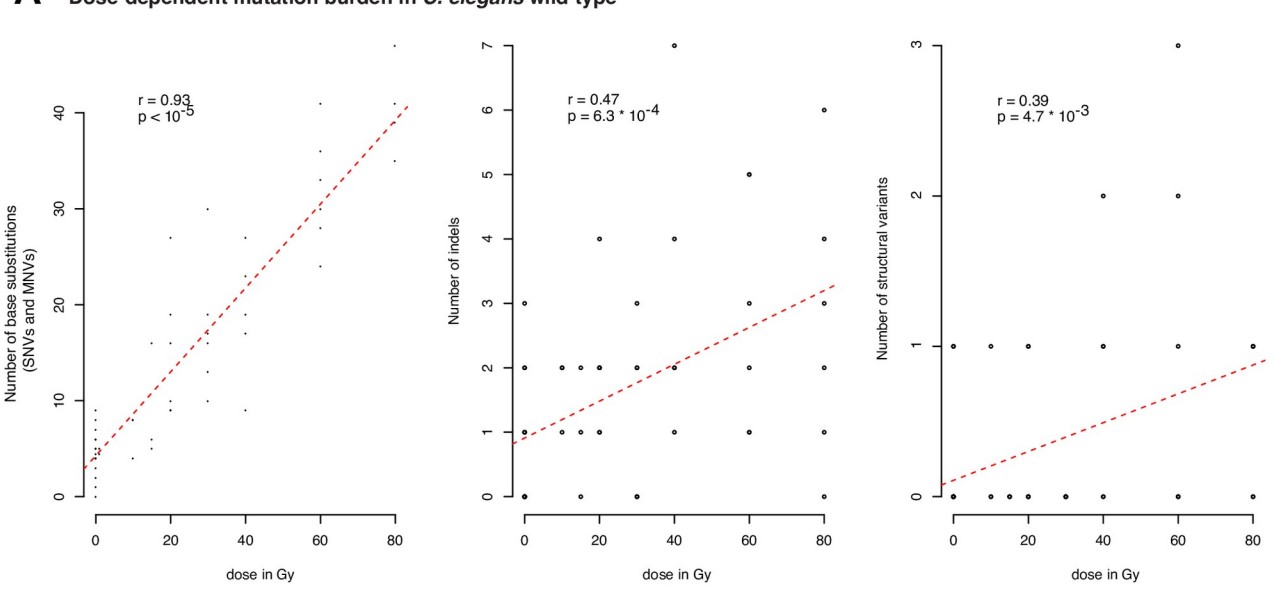

**B** γ-radiation induced mutation rates in *C.elegans* wild-type and DNA repair mutants

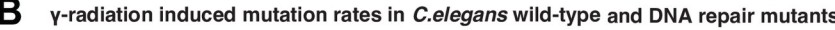
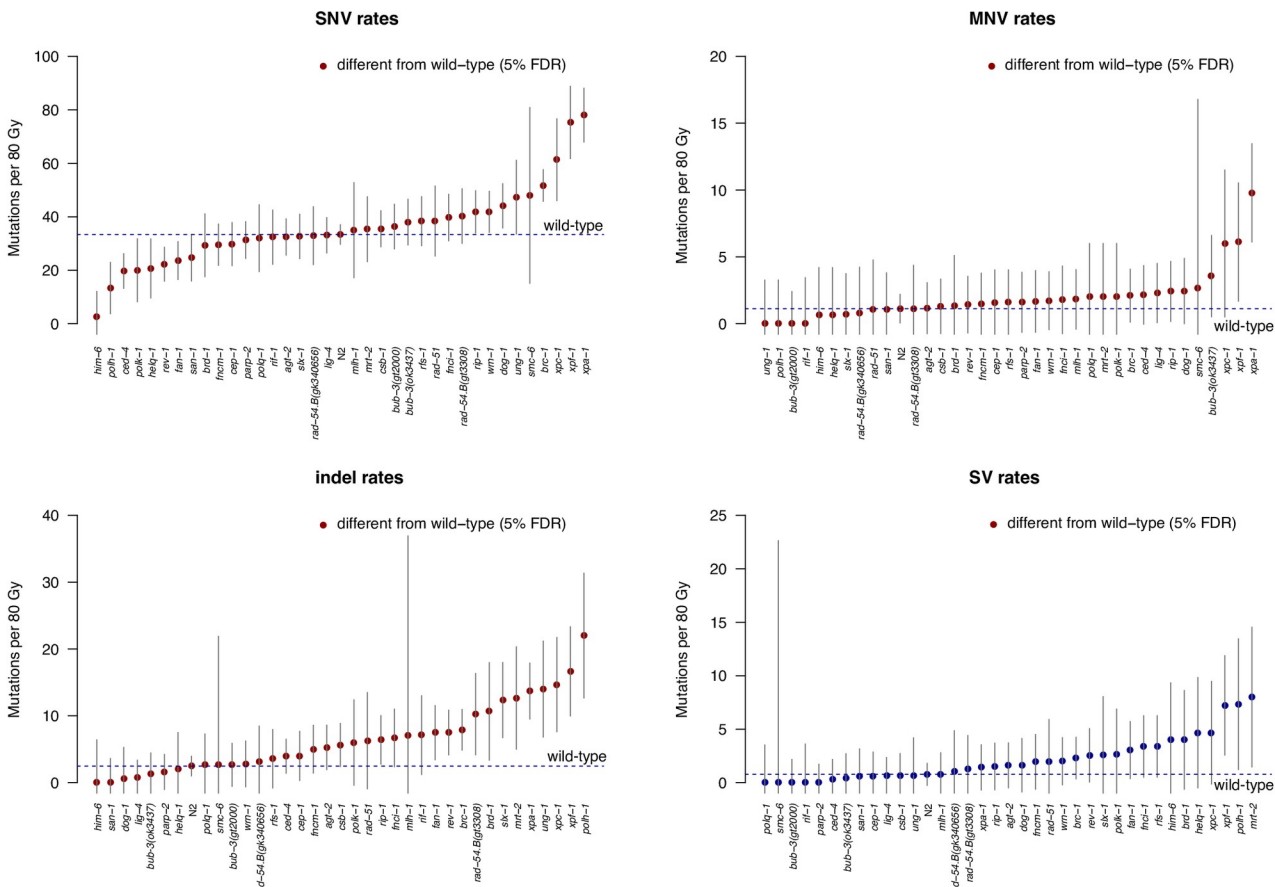

**Fig 2. Dose dependent mutation burden in wild-type and DNA repair mutants. A.** Number of mutations observed across Cs-137-irradiated wild-type samples by dose and mutation type; single (SNVs) and multi- nucleotide variants (MNVs) (left panel), indels (center panel) and structural variants (left panel). Black dots represent mutations observed in individual samples at a given dose, red lines represent a best fit linear regression. The values of **r**

and **p** denote the Pearson correlation coefficient and its two-sided p-value between the number of mutations and radiation dose for each mutation type, respectively. **B.** Numbers of IR-induced mutations in DNA repair mutants compared to wild-type (dashed line); single nucleotide variants (SNVs) (top left), multi-nucleotide variants (MNVs) (top right), indels (bottom left) and structural variants (SVs) (bottom right). Blue and red dots indicate the average number of mutations per 80 Gy IR; red dots represent samples that exhibit statistically significantly different mutation numbers to wild-type (FDR adjusted p-value < 0.05). Grey bars denote 95% confidence intervals.

base uracil glycosylase. This enzyme facilitates base excision repair by generating an abasic site upon uracil removal, uracil arising from the deamination of cytosine (Fig 2B and S2 File and S4 Fig). Finally, the number of indels was increased in *rev-1* and *polh-1* translesion synthesis mutants (Figs 2B and 3B and S2 File and S5 Fig). None of the mutants examined showed a statistically increased rate of SVs.

Of the direct DNA repair pathways, BER and SSBR, mutants of *agt-2* encoding for a putative alkyl-guanine DNA methyltransferase and *parp-2*, encoding for a Poly ADP ribose polymerase did not show altered rates of mutagenesis (S4 Fig). *agt-1*, *apn-1*, *exo-1*, *ndx-1*, *parp-1* and *tdpo-1* BER and SSBR mutants were not included into our stringent dataset. While base substitutions tended to be increased in these lines similar to wild-type irradiated with 40 Gy, no further dose response was observed at 80 Gy possibly due to an experimental error (S4 Fig). All in all, our data indicate that, except for *ung-1* mutants, which showed a higher rate of indels, BER and SSBR mutants did not exhibit IR-induced mutagenesis above wild-type levels.

## DNA repair defects associated with decreased levels of SNVs

Interestingly, *him-6* mutants, defective for the *C. elegans* homolog of the BLM helicase involved in multiple steps of HR (Figs 2B and S4B) [37], *polh-1* and *rev-1* translesion polymerase mutants (Figs 2B and S5), as well as *ced-4* mutants, defective for the *C. elegans* Apaf-1 like apoptosis gene, showed a reduced number of SNVs upon IR exposure (Figs 2B and S6). The evidence of decreased SNVs in *him-6* is supported by dose-dependent wild-type levels of indels and SVs (Figs 2B and S4). POLH-1 translesion synthesis polymerase mutants were unique, showing a significantly altered mutational profile (Fig 2C) with reduced number of SNVs concomitant with increased numbers of indels (Figs 2C and 3 and S2 File and S5 Fig). Reduced SNV numbers were also observed in *polh-1* and *rev-3* mutants treated with alkylating agents, in line with the role of these polymerases in error-prone bypass of damaged bases [26].

## Dinucleotide variants

Dinucleotide substitutions or variants (DNVs) occur more frequently in cancers than expected from calculating random SNV distributions, and thus likely reflect distinct mutational processes [35]. Investigating the contribution of DNA repair pathways to Cs-137-induced DNVs, we found that DNVs were observed infrequently in wild-type upon Cs-137 treatment (Figs 1B and 2B and 3 and 3B and S2 and S3). Among DNA repair mutants tested, DNVs were most prevalent in *xpa-1*, *xpc-1* and *xpf-1* NER and *brc-1* HR mutants (Figs 2B and 4B), indicating that NER and HR are required to repair DNA lesions that otherwise lead to dinucleotide substitutions. Interestingly, *csb-1* mutants, specifically defective for transcription-coupled NER, did not show increased levels of DNVs (Figs 2B and 4B), suggesting that GG-NER plays a major role in preventing dinucleotide substitutions upon IR treatment. The pattern of IR-induced DNVs in *C. elegans* is not closely related to any DNV pattern deduced from cancer genomes, including DNV signatures DBS2 and DBS11, both of which are largely composed of CC>NN changes [35] (Fig 4B, lower panels).

In *C. elegans* DNVs arise from IR, UV and cisplatin exposure, however on different bases and with different mutation outcomes (Fig 4A). UV exposure predominantly led to CC>AT

**A** Similarity of mutation profiles in DNA repair mutants to the IR signature without interactions

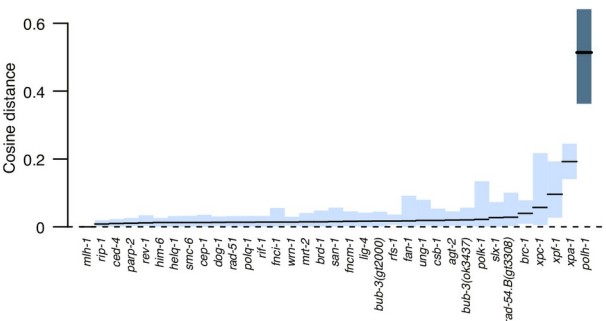

**B** IR-induced mutational signatures in wild-type and selected DNA repair mutants

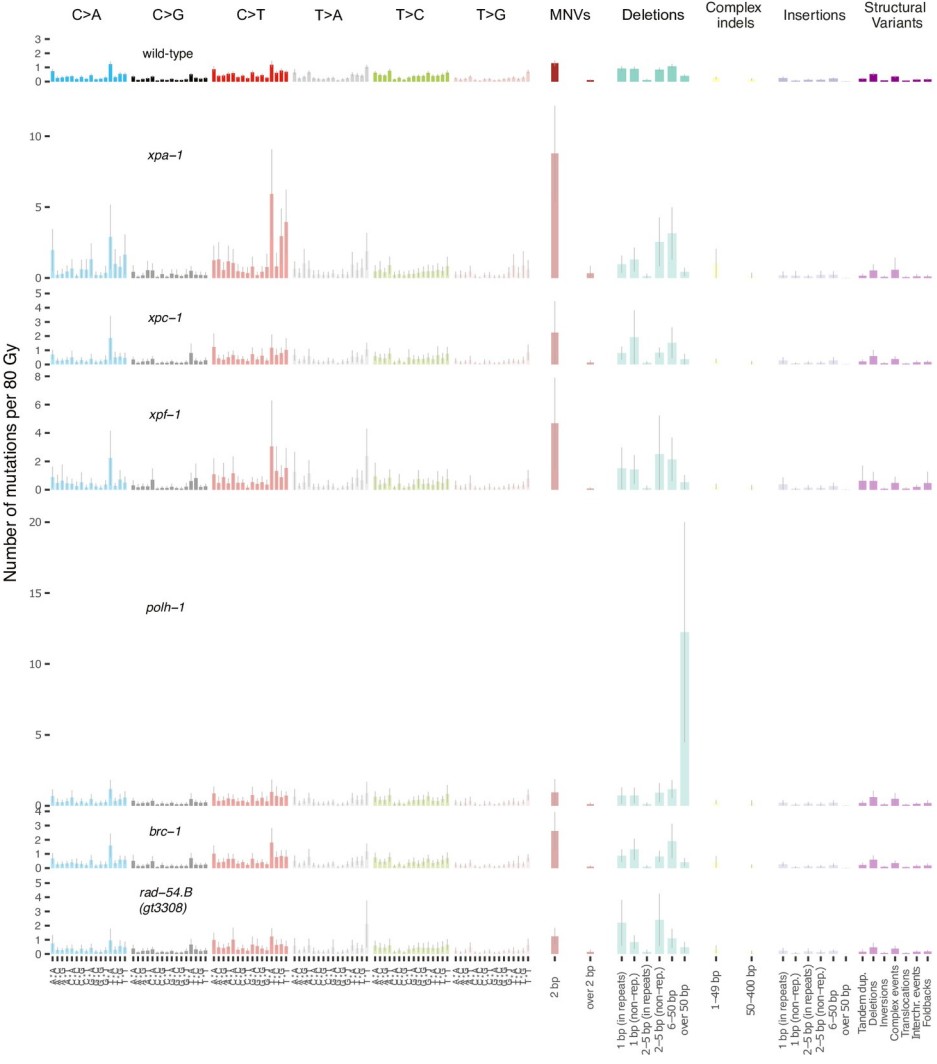

**Fig 3. IR-induced mutation profiles of DNA repair mutants significantly different from wild-type. A.** Similarity of IR-induced mutation profiles in DNA repair mutants compared to wild-type. The level of divergence from the wild-

type mutation profile is indicated by cosine distance. Black bars represent mean cosine distances, blue shaded areas confidence intervals. Dark blue shaded bars indicate mutation profiles which are significantly different from wild-type. **B.** Mutation profiles that are different from wild-type as determined in panel A. Each barplot represents the number of Cs-137 radiation induced mutations per average dose of 80 Gy. Coloured bars represent the mean number of mutations per mutation type, black bars indicate 95% confidence intervals.

and CC>TT changes (Fig 4A), the relative proportion being increased in *xpf-1* mutants (Fig 4A and 4C). CC>TT changes are the hallmark of cancer DNV signature DBS11, associated with UV-induced melanoma [35]. Cisplatin exposure, predominantly led to CT>AC changes [27] (Fig 4D) a signature related to the cancer DNV signature DBS5, associated with cisplatin-induced tumours and characterised by CT>AA and CT>AC changes [35]. In summary, our data shows that dinucleotide variants arise following a variety of DNA insults. This is consistent with the high propensity of dinucleotide substitutions in a variety of human cancer types. DNVs likely occur during the repair of DNA crosslinks between adjacent bases, DNA lesions that are recognized and can be repaired error-free by the NER pathway.

## IR-induced mutation clusters

Both the direct action of IR, which can induce base damage and breaks in the DNA backbone, as well as the indirect action via reactive oxygen species has the potential to create clustered DNA damage. Furthermore, DNA repair synthesis in HR and DSB repair by non-homologous end-joining or microhomology-mediated end-joining are error-prone, potentially resulting in clustered mutations [23–25]. We thus analysed the clustering of IR-induced mutations. Globally, mutations were randomly distributed across chromosomes throughout all wild-type (Fig 5A and S3 File) and DNA repair deficient lines analysed (S3 File). Analysing local mutation distribution, we observed significant mutation clustering in irradiated wild-type lines compared to untreated wild-type. Overall, 6% of all mutations occurred in clusters ($P < 1.7x10^{-7}$) with ~ 1.2 clusters induced per 80 Gy per genome ($P < 1x10^{-7}$) (Figs 5A and 5B and S7 and S2 Table) Clustering was not observed in unchallenged *C. elegans* strains propagated over generations [38]. Clusters typically spanned 10–20 bp and on average consisted of 2–3 mutations (Fig 5C). Interestingly, clustered mutations did not occur more frequently in DNA repair mutants including those deficient in error-free and error-prone DSBR (Figs 5A–5C and S7). Sequence analysis of 12 radiation-associated secondary malignancies from patients with different tumour types was recently reported [17]. Using the same primary data, we observed 10–150 mutation clusters in each of these genomes with 1–10% of all mutations occurring in clusters (S8A and S8B Fig). As in *C. elegans*, most clusters spanned less than 50 bp and encompassed predominantly 2 and rarely more than 4 bases (S8C and S8D Fig). However, the frequency and type of mutation clusters were not significantly different between radiation-induced cancers and cancers of the same type not associated with a history of radiation exposure (S8E and S8F Fig). Thus, these data do not provide firm evidence for mutational clustering linked to IR exposure in human secondary malignancies.

## Discussion

Using *C. elegan*s as a model, our study is the first to systematically describe IR-induced mutation rates and patterns in wild-type and DNA repair and damage response defective strains encompassing the vast majority of known pathways needed for mending DNA lesions. In line with expectations of IR induced mutation rate estimates in mouse and *C. elegans* germ cells [8, 9, 39] and from sequencing the progeny of irradiated mice, budding yeast and several plant species (for instance [10–16]), we confirmed that IR-induced mutagenesis is surprisingly low,

**A**  Genotoxin-induced dinucleotide substitution profiles

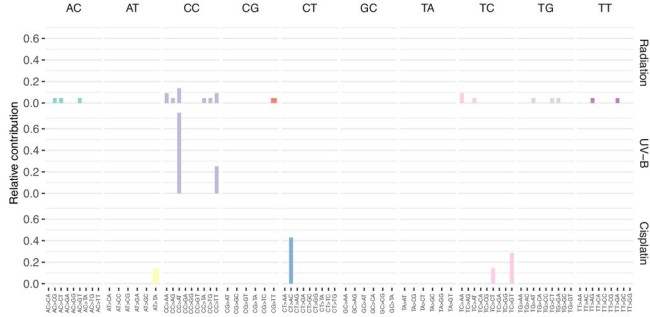

**B**  Cs137 radiation-induced dinucleotide substitutions in *C. elegans* DNA repair mutants compared to dinucleotide substitution profiles observed in cancer

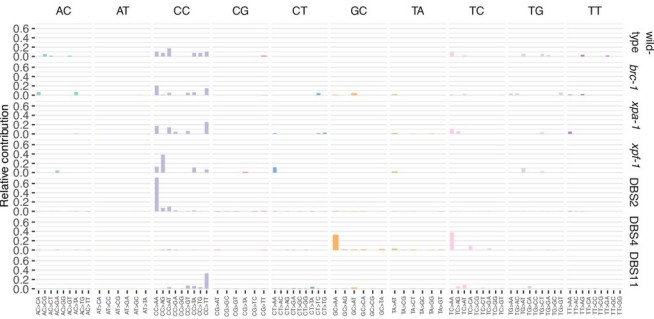

**C**  UV-induced dinucleotide substitutions in *C. elegans* DNA repair mutants compared to dinucleotide substitution profiles observed in cancer

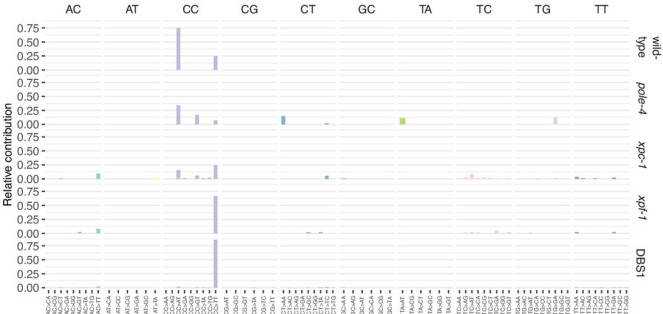

**D**  Cisplatin-induced dinucleotide substitutions in *C. elegans* DNA repair mutants compared to dinucleotide substitution profiles observed in cancer

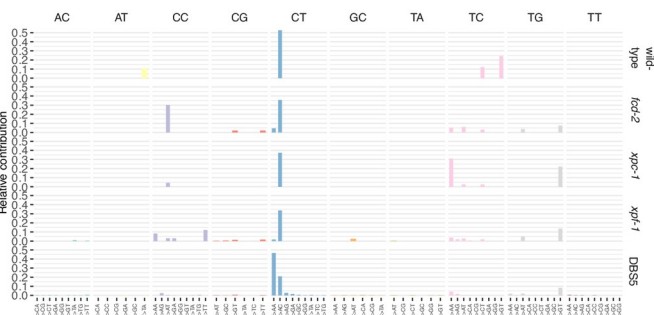

**Fig 4. Dinucleotide substitution profiles induced by ionising radiation, UV-light and cisplatin exposure in *C. elegans* and human cancers. A.** Relative contribution of dinucleotide substitutions observed in *C. elegans* wild-type following Cs-137, UV-B, and cisplatin exposure. **B.** Cs-137-induced dinucleotide substitutions in *C. elegans* wild-type and DNA repair mutants compared to closely related COSMIC dinucleotide substitution signatures described in human cancers. **C.** UV-induced dinucleotide substitutions in *C. elegans* wild-type and DNA repair mutants compared to closely related COSMIC dinucleotide substitution signatures described in human cancers. **D.** Cisplatin-induced dinucleotide substitutions in *C. elegans* wild-type and DNA repair mutants compared to a closely related COSMIC dinucleotide substitution signature described in human cancers.

with 80 Gy inducing an average of 36.6 SNVs, 1.3 DNVs, 4 indels and 1.4 (SD = 0.13) SVs in the 100 million base *C. elegans* genome. The highest proportion of mutants being SNVs is in line with the analysis of IR induced secondary tumours [17]. In contrast, increased SNVs could not be detected in thyroid carcinomas, linked to the Chernobyl disaster [19] and irradiated human organoids [20].

We typically did experiments in triplicates, assessing samples without, and with an intermediate and high dose of IR. Clearly, increased statistical power would require the analysis of a larger number of samples. It is impossible to significantly increase the dose of IR as this would lead to 100% embryonic lethality. Broadly, 80 Gy leads to a 30 to 50% reduction of survival of irradiated progeny, with up to 10-fold lower survival in *C. elegans* DNA repair mutants [40–45]. Radiation sensitivity inversely correlates with genome size: Only 8–12 Gy are needed to cause 50% lethality in mice 5–8 weeks after whole body irradiation [46], mice having a 25 times larger genome than *C. elegans* with its 100 million base haploid genome. In contrast, ~1000 Gy are needed for 50% lethality in diploid budding yeast cells carrying a 12 million base haploid genome, haploid cells being ~20 times more sensitive [47]. Assuming that SVs are the most deleterious mutations, it appears unlikely that the reduced survival can solely be ascribed to IR-induced DNA lesions. This hypothesis is supported by the presence of predominantly heterozygous SVs detected in our experimental system; heterozygous SVs generally not causing phenotypes (also see discussion below) (Fig 1A). Also, it is unlikely us having missed complex chromosomal rearrangements, us having detected those previously in *C. elegans* lines treated with the DNA crosslinking agents cisplatin and nitrogen mustard using the same analysis pipeline [27]. On the other hand, while a large number of *C. elegans* fusion chromosomes are known to be viable in a heterozygous state and are being used as balancer chromosomes [48], aneuploidy of autosomes, typically caused by meiotic segregation defects, leads to embryonic lethality [49]. It is thus possible that we fail to detect a subset of chromosomal fusions or the loss of large chromosomal fragments resulting from unrepaired DSBs, if these events lead to a near aneuploid state and consequently to embryonic or larval lethality. In other words, we might fail to detect highly mutagenised genomes because affected embryos die, and do not sire progeny needed for whole genome sequence analysis. Detection of mutational events leading to embryonic lethality would require single embryo or single zygote whole genome sequencing, studies we will pursue in the future.

When the consequence of germline mutagenesis is scored in the subsequent generation, it is important to consider the nature of the germ cells exposed to IR, mindful that mutagenic outcomes of individual germ cell cells are scored. Such a quantal approach likely explains the relatively large variation we observe in parallel experiments (Fig 1B and S1 File). Our experimental system, which mimics the procedure used in mutagenesis *C. elegans* genetic screens over the past six decades [9], takes advantage of *C. elegans'* self-fertilizing reproduction to clonally amplify the genome from single zygotes, derived from male and female germ cells treated with IR (Fig 1B). Based on the timing of germline differentiation, including the staging of mitotic and meiotic cell cycles, the timing after IR treatment is chosen such that female germ cells are irradiated in meiotic pachytene. In pachytene, the initiation of meiotic recombination

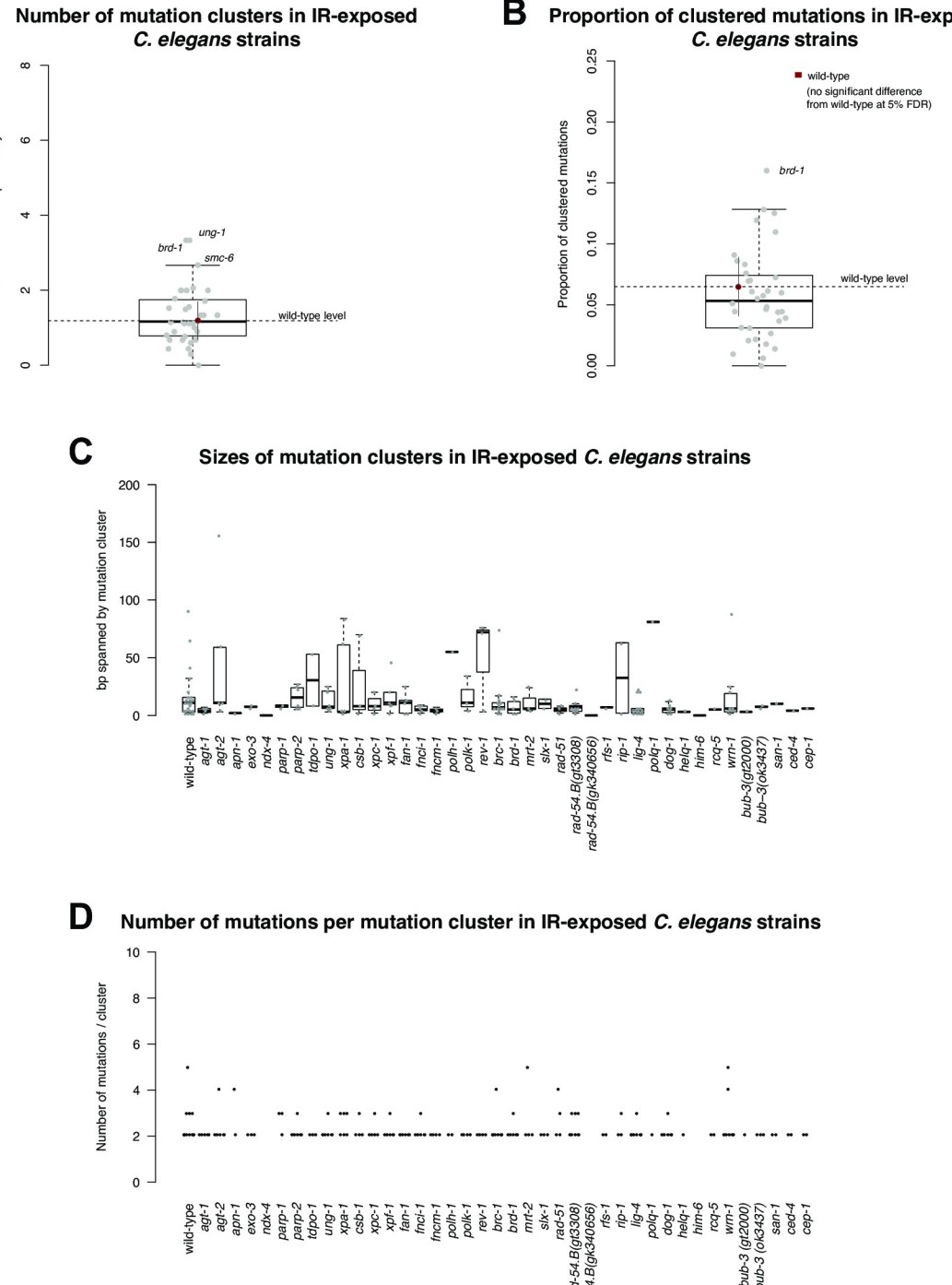

**Fig 5. IR-induced mutation clusters. A.** Number of mutation clusters in IR-exposed *C. elegans* strains per 80 Gy of Cs-137 radiation. Grey dots indicate the number of mutation clusters in different *C. elegans* mutants. The red dot and dashed line indicate the number of mutation clusters observed in wild-type. Black bars indicate the median mutation cluster size, squares the interquartile range and the error bar the low and high 1.5*interquartile ranges. **B.** Proportion of IR-induced mutations within clusters for wild-type (red dot, dashed line) and DNA repair deficient *C. elegans* lines (grey dots) as depicted in A. **C.** Boxplots representing the sizes of IR-induced mutation clusters in bp for wild-type and DNA repair mutants. Grey dots represent the size of individual mutation clusters observed for the respective genotype. **D.** Number of mutations per mutation cluster (black dots) in IR-exposed wild-type and DNA repair mutants.

is completed and maternal and paternal chromosomes tightly align into the so-called synaptonemal complex [49, 50]. Male germ cells are produced in *C. elegans* hermaphrodites before female germ cells are formed and are stored in the spermatheca after the completion of meiotic divisions [51]. It remains to be investigated if mutation rates differ between male and female or between mitotic and meiotic germ cells. A meiotic checkpoint, which only acts on female germ cells, leads to the elimination of a large proportion of female pachytene cells in *C. elegans* by apoptosis [40]. Apoptosis is compromised in *ced-4* mutants, defective for the *C. elegans* Apaf-1 like gene [40, 52]. CEP-1, the sole p53-like gene [53–55] is essential for DNA damage induced apoptosis triggered by IR [40]. Surprisingly, we did not observe increased mutagenesis in either mutant. Thus, our analysis does not provide evidence that apoptosis is specifically used to eliminate germ cells subjected to excessive DNA damage.

Our data supports previous evidence that only a minority of DNA double-strand breaks inflicted by ionizing radiation is converted into more severe genomic changes. In mammalian cells, about 40 DNA double-strand breaks are induced per Gy of IR [24, 25]. In *C. elegans*, antibody-stained RAD-51 recombinase foci indicate that ~26 DSBs are present in mitotic germ cells 6 hours after treatment with 120 Gy [41]. About 10 times more DSBs are induced in meiotic cells (24/10 Gy), as estimated by the dose of IR required to bypass a defect in the SPO-11 nuclease essential for meiotic DSB formation [56]. All in all, these results indicate that the vast majority of DNA lesions, including DNA double-strand breaks which can lead to SVs such as insertion, deletions, duplications, and translocations are repaired by various DNA repair pathways. It appears surprising that mutations defective in DNA end-joining (EJ) and microhomology mediated end-joining (MMEJ) do not exhibit increased IR-induced mutagenesis. However, these results are consistent with EJ's predominant function in somatic cells and a likely redundancy of MMEJ with HR [57–60]. More surprising is our finding that mutations affecting checkpoint signalling or HR, including complete loss of function mutants of *brd-1* and *brc-1* corresponding to human BRCA1 and BARD1 genes, as well as *him-6*, the *C. elegans* BLM helicase homolog, do not show increased numbers of SVs. In some of these mutants, numbers of SVs tended to be increased, but did not reach statistical significance (Fig 2). Importantly, HR is essential for meiotic recombination. However, *brc-1*, *brd-1* and *him-6* null mutants are viable and only partially compromised for meiotic HR suggesting genetic redundancy in HR. Nevertheless, *brc-1*, *brd-1* and *him-6* deficiencies show reduced survival upon exposure to IR [37, 61]. Also, these three HR mutants show increased numbers of SVs when propagated over 40 generations in the absence of genotoxin exposure [62]. Finally, *brc-1*, *brd-1*, and *slx-1*, a nuclease involved in HR intermediate resolution, [63, 64], show increased numbers of indels, which might also be products of error-prone DSB repair. All in all, these data indicate that DSB repair pathways are highly redundant and that redundant mechanisms also act during HR.

We were intrigued to observe that the number of IR-induced single nucleotide variants was strongly reduced in mutants defective for *him-6*, even more so than in *polh-1* translesion synthesis polymerase mutants. Reduced SNV numbers in translesion synthesis mutants suggest that a sizable proportion of SNVs are generated by POLH-1 likely by incorporation of incorrect nucleotides opposite damaged bases. The failure to bypass damaged bases in the absence of POLH-1 can explain the observed increase in small deletions, which are also present in *polh-1* mutants treated with a variety of alkylating agents, UV, and the bulky DNA adduct forming agents aflatoxin B1 and aristolochic acid [26]. At present, we can only speculate why less SNVs are present in *him-6* BLM mutants. The late stages of HR are highly redundant and two major pathways, Holliday Junction dissolution mediated by BLM and Holliday Junction resolution mediated by resolving enzymes, such as the GEN-1 nuclease or the SLX-4/SLX-1/ MUS-81 nuclease complex, are needed to resolve double Holliday Junctions, that covalently

link resected DNA double-strand breaks with template DNA [65–67]. HR is also used to mend DNA base damage that halts DNA replication. Such a situation can lead to fork reversal, the formation of a 'chicken foot structure' [68–70], and the subsequent error-free bypass of the damaged base through recombinogenic pathways, which likely involves GEN1 nuclease and/ or the SLX-4/SLX-1/MUS-81 resolvases. It is possible that in the absence of HIM-6 more lesions are funnelled into the error-free GEN1 and SLX-4/SLX-1/MUS-81 dependent pathways. Indeed, the increased level of SNVs we observe in *brc-1* mutants might be indicative of error-prone translesion synthesis, employed when damaged bases are neither repaired by BER or NER nor by recombination linked to replication fork reversal [70].

The increase of SNVs and DNVs in NER mutants indicates an important role of this pathway in preventing single and dinucleotide changes. These results appear surprising at first glance, as base damage inflicted by reactive oxygen species is largely not helix distorting and thus generally thought to be mended by BER, the deficiency of which in our system had only minimal effects. However, radiation induced reactive oxygen species are known to induce bulky cyclodeoxynucleosides by affecting intramolecular crosslinks between the C-8 position of adenine or guanine and the 5' position of 2-deoxyribose, cyclodeoxynucleosides being excised by NER [71, 72]. It is possible that the increased rate of C>T changes (G>A on the other strand) observed in *xpa-1* and *xpf-1* mutants may be related to the failure to repair cyclodeoxyguanine. In addition, ionizing radiation is known to cause multiple types of intrastrand crosslinks [73–77], which may account for radiation-induced dinucleotide changes repaired by NER.

Radiation tracks typically generate more than one reactive oxidative species when penetrating DNA and surrounding water layers [23–25], implying that clustered DNA damage should be highly prominent. While we do observe an increase in clustered mutagenesis, only 6% of all mutations are clustered. Thus, ionizing events in the majority of cases may only cause single mutations or DNA repair of clustered DNA damage is highly efficient and largely error-free. Since we did not find any DNA repair pathway whose inactivation substantially increased clustered mutations, the former possibility seems more likely. Alternatively, the repair of clustered DNA lesions might be highly redundant.

Taken together, we show that IR induces multiple types of mutations. Our data also implies that DNA repair pathways mend the vast majority of DNA lesions caused by IR. Moreover, DNA repair pathways are highly redundant for IR-induced damage repair as single DNA repair mutants, if at all, generally only show a modest increase of mutagenesis. The most dramatic effect was observed in *xpf-1* mutants, likely reflecting the function of the XPF-1 nuclease in both nucleotide excision and DNA double-strand break repair [78, 79] and possibly also in DNA crosslink repair [80, 81]. Our data suggest that drugs targeting this nuclease might be useful to sensitise cancer killing by IR.

## Supporting information

**S1 Table. Strain and sample description.** Details of *C. elegans* strains used in and sequencing samples of this study. For sample description unless stated otherwise primary whole genome sequencing data were deposited (Filtered VCF files) in Volkova, Meier et al., Nat Commun. 2020 (Supplementary Data 6 file collection) [1].
(XLSX)

**S2 Table. Clustered mutations and cluster summary.** Details of all mutations reported as clustered across genotypes and different irradiation doses, as well as a summary of clustering for all samples where any clustered mutations were found.
(XLSX)

**S1 Fig. Comparative analysis of indels and structural variants observed in radiation-associated secondary malignancies and Cs-137-irradiated *C. elegans*. A.** Average number of indels observed in secondary malignancies (1) (top panel) and Cs-137 irradiated *C. elegans* (bottom panel). Indels are classified as deletions (DEL), deletions with insertions (INDEL) and insertions (INS) with indicated size ranges. Error bars represent confidence intervals with 2 standard errors of the mean (SEM) (top panel) and 95% credible intervals (bottom panel). **B.** Average numbers of structural variants observed in secondary malignancies (1) (top panel) and in Cs-137 irradiated *C. elegans* (bottom panel) by type. Error bars represent confidence intervals with 2 standard errors of the mean (SEM) (top panel) and 95% credible intervals (bottom panel).
(TIFF)

**S2 Fig. Mutation numbers and types observed in NER and CLR mutants following irradiation with Cs-137. A.** Mutation numbers by class and type observed in wild-type and nucleotide excision repair (NER) mutants for indicated radiation doses and 3 irradiated lines per genotype. Mutation types are shown as the 6 possible single nucleotide variants, dinucleotide variants (DNV), multi-nucleotide variants (MNVs) (top panel), insertions (INS) and deletions (DEL) and deletions with insertions (INDEL) (centre panel), and structural variants (SVs) encompassing tandem duplications (TD), large deletions (DEL), inversions (INV), translocations (TRSL), foldback (FOLDBACK), interchromosomal (INTCHR) and complex (COMPLEX) SVs (lower panel). **B.** Mutation numbers by class and type observed in wild-type and DNA crosslink repair (CLR) mutants for indicated radiation doses and 3 irradiated lines per genotype. Mutation types as described in A. N.D. indicates lines for which no DNA sequencing information could be obtained. Strains with mutational patterns statistically significantly different from wild-type are indicated by an asterisk '*' (P ≤ 0.05). Strains lacking a dose-response and thus were excluded from our detailed analysis are indicated with ∉.
(TIFF)

**S3 Fig. Mutation numbers and types observed in DSBR and helicase mutants following irradiation with Cs-137. A.** Mutation numbers by class and type observed in wild-type and double-strand break repair (DSBR) mutants for 3 irradiated lines per genotype and indicated radiation dose. **B.** Mutation numbers by class and type observed in wild-type and helicase mutants for indicated radiation doses and 3 irradiated lines per genotype. N.D. indicates lines for which no DNA sequencing information could be obtained. Strains with mutational patterns statistically significantly different from wild-type are indicated by an asterisk '*' (P ≤ 0.05). Strains lacking a dose-response and thus excluded from our detailed analysis are indicated with ∉.
(TIFF)

**S4 Fig. Mutation numbers and types observed in MGMT and BER mutants following irradiation with Cs-137. A.** Mutation numbers by class and type observed in wild-type and putative direct damage reversal repair 06-Methylguanine methyltransferase (MGMT) mutants for indicated radiation doses and 3 irradiated lines per genotype. **B.** Mutation numbers by class and type observed in wild-type and base excision repair (BER) mutants for indicated radiation doses and 3 irradiated lines per genotype. N.D. indicates lines for which no DNA sequencing information could be obtained. Strains with mutational patterns statistically significantly different from wild-type are indicated by an asterisk '*' (P ≤ 0.05). Strains lacking a dose-response thus excluded from our detailed analysis are indicated with ∉.
(TIFF)

**S5 Fig. Mutation numbers and types observed in TLS mutants following irradiation with Cs-137. A.** Mutation numbers by class and type observed in wild-type and translesion synthesis (TLS) mutants for indicated radiation doses and 3 irradiated lines per genotype. N.D. indicates lines for which no DNA sequencing information could be obtained. Strains with mutational patterns statistically significantly different from wild-type are indicated by an asterisk '*' ($P \leq 0.05$). Strains lacking a dose-response thus excluded from our detailed analysis are indicated with $\notin$.
(TIFF)

**S6 Fig. Mutation numbers and types observed in SAC and apoptosis mutants following irradiation with Cs-137. A.** Mutation numbers by class and type observed in wild-type and spindle assembly checkpoint (SAC) mutants for indicated radiation doses and 3 irradiated lines per genotype. **B.** Mutation numbers by class and type observed in wild-type and apoptosis mutants for indicated radiation doses and 3 irradiated lines per genotype. N.D. indicates lines for which no DNA sequencing information could be obtained. Strains with mutational patterns statistically significantly different from wild-type are indicated by an asterisk '*' ($P \leq 0.05$). Strains lacking a dose-response thus excluded from our detailed analysis are indicated with $\notin$.
(TIFF)

**S7 Fig. IR-induced mutation clusters in DR, BER and RecQ helicase mutants.** Sizes of IR-induced mutation clusters shown in bp spanned by the mutation cluster (top panel) and as the number of mutations per cluster (bottom panel). Grey or black dots indicate bp spanned or mutations by cluster, respectively, observed in individual lines of the indicated genotypes. Black bars of the boxplot (right top panel) indicate the median mutation cluster size, squares the interquartile range and error bars 1.5*interquartile ranges.
(TIFF)

**S8 Fig. Mutation clusters in human tumours with and without association to IR-exposure. A.** Observed number of mutation clusters in 12 IR-associated human secondary malignancies (Material and Methods) [1]. **B.** Proportion of clustered mutations in 12 IR-associated human secondary malignancies. **C.** Boxplots depicting the sizes of mutation clusters in 12 IR-associated human secondary malignancies in bp. Black bars indicate the median mutation cluster size, squares the interquartile range, and error bars 1.5* interquartile ranges. Grey dots represent the size of individual mutation clusters observed in the respective tumour. **D.** Number of mutations per mutation cluster in 12 IR-associated human secondary malignancies. Black dots represent the number of mutations observed in individual clusters of the respective tumour. **E.** Number of mutation clusters in different human cancers. Mutation clusters are shown as the relative number of mutation clusters to total mutation burden. Grey dots represent the number of mutation clusters in 33 individual non-IR associated BRCA negative breast cancers, 3 IR-associated secondary breast cancers (PD8618a, PD8622a, PD8623a, Panel A-D), 62 non-IR associated bone cancers, and 9 IR-associated bone cancers including angio-, osteo-, and spindle cells sarcomas (Panel A-D). Outliers are highlighted by a black outline. Black bars indicate the median mutation cluster size, squares the interquartile range and error bars 1.5* interquartile ranges. **F.** Proportion of clustered mutations in different human cancers shown in non-IR and IR-associated human cancers as described in E.
(TIFF)

**S1 File. Wild-type and DNA repair mutant dose response.** Details of the observed number of mutations in all samples of *C. elegans* wild-type and mutant strains across radiation doses. Number of mutations observed across all Cs-137-irradiated samples by dose and mutation

type: single nucleotide variants (left panel), indels (center panel) and structural variants (left panel). Black dots represent mutations observed in individual samples at a given dose, red lines represent a best fit linear regression.
(PDF)

**S2 File. IR-induced mutation profiles of wild-type and DNA repair mutants.** Each barplot reflects the number of mutations per average dose of 80 Gy of Cs-137-radiation. Three stars indicate samples with SNVs, MNV, indels, or SV significantly different to wild-type (FDR < 5%). Bold lines below a mutation class indicate that a specific substitution type with the above classes differs from wild-type (FDR < 5%).
(PDF)

**S3 File. Chromosomal location of mutations in wild-type and DNA repair mutants.** Chromosomal location of mutations observed after exposure to the indicated radiation doses across the 5 *C. elegans* autosomes (I-V) and the X chromosome for all genotypes. Small circles, triangles, and squares, indicate single SNVs, indels, and DNVs, respectively. Larger circles, triangles, and squares, indicate clustered SNVs, indels, and DNVs, respectively.
(PDF)

**S4 File.**
(DOCX)

## Acknowledgments

We thank Orlando D. Schärer for helpful discussions on the NER pathway and Dmitry Ivanov for proofreading. We are grateful to the Mitani Lab (National Bio-Resource Project, Japan), and the *Caenorhabditis* Genetics Center (P40 OD010440) for providing strains.

## Author Contributions

**Conceptualization:** Peter J. Campbell, Moritz Gerstung.

**Data curation:** Nadezda V. Volkova, Moritz Gerstung.

**Formal analysis:** Bettina Meier, Nadezda V. Volkova, Moritz Gerstung, Anton Gartner.

**Funding acquisition:** Peter J. Campbell, Moritz Gerstung, Anton Gartner.

**Investigation:** Nadezda V. Volkova, Víctor González-Huici, Moritz Gerstung, Anton Gartner.

**Methodology:** Moritz Gerstung, Anton Gartner.

**Project administration:** Anton Gartner.

**Resources:** Bin Wang, Víctor González-Huici, Simone Bertolini, Moritz Gerstung.

**Software:** Bettina Meier, Nadezda V. Volkova.

**Supervision:** Peter J. Campbell, Moritz Gerstung, Anton Gartner.

**Visualization:** Bettina Meier, Nadezda V. Volkova, Moritz Gerstung.

**Writing – original draft:** Bettina Meier, Nadezda V. Volkova, Moritz Gerstung, Anton Gartner.

**Writing – review & editing:** Bettina Meier, Nadezda V. Volkova, Moritz Gerstung, Anton Gartner.

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
