## [Decision Letter · Decision Letter 0]

21 Jul 2021

PONE-D-21-18200

C. elegans genome-wide analysis reveals DNA repair pathways that act cooperatively to preserve genome integrity upon ionizing radiation

PLOS ONE

Dear Dr. Gartner,

Thank you for submitting your manuscript to PLOS ONE. After careful consideration, we feel that it has merit but requires some  adjustments to fully meet PLOS ONE’s publication criteria. These are discussed in detail in the 2 sets of reviewers comments attached with this letter.Therefore, we invite you to submit a revised version of the manuscript that addresses the points raised.

We look forward to receiving your revised manuscript.

Kind regards,

Sue Cotterill

Academic Editor

PLOS ONE

Journal Requirements:

2. We noted in your submission details that a portion of your manuscript may have been presented or published elsewhere. [We note that our current manuscript uses the same primary sequence source data we deposited as part of our Nat Commun. 2020 May 1;11(1):2169. doi: 10.1038/s41467-020-15912-7. We clearly indicate this in the abstract, the results section, and the discussion. In the Nature Communication paper, we provided a high-level overview (and comparison) of the mutagenic consequences of exposure to 11 genotoxic agents, one agent being ionizing radiation (IR).] Please clarify whether this publication was peer-reviewed and formally published. If this work was previously peer-reviewed and published, in the cover letter please provide the reason that this work does not constitute dual publication and should be included in the current manuscript.

Additional Editor Comments:

Reviewers' comments:

Reviewer's Responses to Questions

**Comments to the Author**

1. Is the manuscript technically sound, and do the data support the conclusions?

Reviewer #1: Yes

Reviewer #2: Yes

2. Has the statistical analysis been performed appropriately and rigorously? 

Reviewer #1: Yes

Reviewer #2: Yes

3. Have the authors made all data underlying the findings in their manuscript fully available?

Reviewer #1: Yes

Reviewer #2: Yes

4. Is the manuscript presented in an intelligible fashion and written in standard English?

Reviewer #1: Yes

Reviewer #2: Yes

5. Review Comments to the Author

Reviewer #1: This study provides interesting observations about the effect of ionizing radiation (IR) on mutational signatures, with respect to different DNA repair and damage response defects, in a model organism C. elegans. Although the sequencing data and some general observations have been already published by the authors as part of a broader study (in Nature Communications), in this study, the authors focus only on the IR-treated samples and more in-depth analysis of the mutational signatures in these samples. The results support a role of GG-NER pathway (XPF-1, XPA-1, and XPC-1) in error-free repair of IR-induced damage and the role of TLS (POLH-1 and REV-1) in error-prone repair of IR-induced damage.

Comments

1. The dose-dependent increase should have statistics performed and reported (such as the correlation coefficient and p-value).

2. “Structural variants (SVs) were augmented with dose”: Was this statistically significant? (The numbers look very low, so maybe the sentence in abstract should be adjusted accordingly.)

3. In the cluster analysis, did you observe any dose effect?

4. “In C. elegans, ~6 % of all IR-induced mutations are clustered, an effect generally not observed in unchallenged C. elegans strains propagated over generations”: Can you make a direct comparison? What proportion of mutations are clustered in unchallenged strains?

5. The link of the github repository refers to a different project.

6. Could you please share the VCF files with the readers? This will simplify reproducibility and potential future use of the data set in the community. It those files are already available in your previous publication, could you please make a simple list of how to find the VCF files relevant for this publication?

7. Would it be possible to include in the github repository everything needed to easily reproduce (at least the main) manuscript figures? (E.g., it seems the scripts expect various input files and parameterisation files, but I did not find those files in the repository.)

8. The methods should be ideally summarized in the methods/supplementary text of this paper in order to enable reproducibility without the need to look for many details and parameterization described in the previous paper. In particular, it would be good if the authors could describe in more detail how the variant filtering against a panel of unrelated samples was performed, as well as the cut-offs on coverage, read support of the variant in the test and control samples, and overlap with other genomic variants.

9. Could you please (in the discussion) briefly comment of similarities and differences of your results to those of mutational signatures of ionizing radiation observed in the following publications:

o (Li et al., 2020) https://www.nature.com/articles/s41467-019-14261-4

o (Kageyama et al., 2020) https://www.thegreenjournal.com/article/S0167-8140(20)30856-2/fulltext

o (Youk et al., 2021) https://www.biorxiv.org/content/10.1101/2021.01.12.426324v1.full

Minor comments

1. Top of the page 10: the formula has some missing characters.

2. Page 13: incorrect font.

3. Page 14: a typo in “A Indeles”.

4. Page 14: “Finally the proportion of indels was increased in rev-1 and pol-h translesion synthesis defective strains”: Why proportion? (The figure seems to show mutation counts.)

Reviewer #2: In this study, authors irradiate the germ cells of C.elegans in wild-type and DNA repair or damage response mutant animals and perform whole genome sequencing on the progeny which have undergone clonal expansion of mutations from zygotes. The study is methodologically sound and the analysis that was performed to describe the nature of these mutations generated by ionizing radiation (IR) is thorough. This is certainly an important and comprehensive study and the data is a valuable resource to the genomics community. While overall it is surprising that the radiation does not have a more substantial effect on mutagenesis in C.elegans, some interesting observations were made. The data suggests that global NER is important for repair of IR induced damage and that error prone bypass of IR lesions by translesion synthesis polymerases cause SNVs. A specific dinucleotide substitution pattern of mutations from irradiation was also identified. Minor comments are made below.

Minor

1. As the human genome is ~30x larger than the C.elegans genome, 3 replicates per parent might not result in enough mutations to make conclusions for lower frequency mutations such as SVs. This possibility should be added to the discussion.

2. Since irradiation affects survival of progeny, can you discuss the possibility that the model system selects for progeny that accumulated less mutations? In the discussion on page 20, authors say this is not the case as apoptosis compromised ced-4 mutants do not have increased mutation burden. Can you preclude other possibilities such as necrosis or loss of capacity for self-fertilization?

3. Figure 1C – Since COSMIC signature 3 and SBS40 are non-distinct signatures, the cosine similarity may appear similar to that of the humanized C.elegans irradiation signature without a genuine biological relationship between the signatures. You should compare the cosine similarity of simulated mutations to signature 3 and 40 and see if the value is lower.

4. Figure 2A – Add p-value and R-squared for line of best fit linear regression. It does not look like irradiation significantly induces indels and SNVs.

5. Figure 2B – The figure legend describes values as fold change but the y-axis is number of mutations. Can you add ‘wild-type level’ on the dotted line (like in figures 5A and 5B) to make the comparison to wild-type clearer.

6. Figure 5 – Can you perform simulations or bootstrapping analysis to see what proportion of mutations would fall in clusters by chance such as randomly shuffling mutations in trinucleotide preserved manner? Or simply, do non-irradiated animals also display mutation clusters?

7. Figure 5 – Where in the genome do mutation clusters lie? For example, are they associated with heterochromatin or other epigenetic marks? Were there any genomic sites of mutation clustering that occurred in more than 1 sample tested.

6. PLOS authors have the option to publish the peer review history of their article (what does this mean?). If published, this will include your full peer review and any attached files.

Reviewer #1: No

Reviewer #2: **Yes: **Jayne Barbour

---

## [Author Response · Author response to Decision Letter 0]

1 Sep 2021

(same letter as ‘coverletter’)

Dear Mrs Cotteril,

We herewith submit our revised manuscript ‘PONE-D-21-18200 ‘C. elegans genome-wide analysis reveals DNA repair pathways that act cooperatively to preserve genome integrity upon ionizing radiation’. 

We are grateful for the support and the helpful comments of the two reviewers. We addressed all of their comments; thereby improving our manuscript. Below, A) please find point to point response to the reviewers. Further below, B) as requested (Journal requirement 2) we provide a detailed case for our manuscript not constituting dual publication; it being distinct from our previous publication. 

All submitted files are in the correct format, and captions for Supporting Information files are provided. 

We are confident that the manuscript is now ready for publication.

best regards,

Anton Gartner and Moritz Gerstung

A) Response to reviewers:

Reviewer #1: This study provides interesting observations about the effect of ionizing radiation (IR) on mutational signatures, with respect to different DNA repair and damage response defects, in a model organism C. elegans. Although the sequencing data and some general observations have been already published by the authors as part of a broader study (in Nature Communications), in this study, the authors focus only on the IR-treated samples and more in-depth analysis of the mutational signatures in these samples. The results support a role of GG-NER pathway (XPF-1, XPA-1, and XPC-1) in error-free repair of IR-induced damage and the role of TLS (POLH-1 and REV-1) in error-prone repair of IR-induced damage.

Comments

1. The dose-dependent increase should have statistics performed and reported (such as the correlation coefficient and p-value).

Answer: Following this request, we have added the correlation coefficient and p-value to Figure 2A to report the statistics behind the dose-dependent effect in wild-type. The mutational effects of ionising radiation in other genotypes (Figure 2B) are reported with their average values, CIs and p-values all in comparison to the wild-type effect.

2. “Structural variants (SVs) were augmented with dose”: Was this statistically significant? (The numbers look very low, so maybe the sentence in the abstract should be adjusted accordingly.)

Answer: As reported now in Figure 2A, the correlation between the number of SVs and irradiation dose was weaker than for SNVs, but nevertheless statistically significant.

3. In the cluster analysis, did you observe any dose effect?

Answer: We did observe a weak dose-dependent effect in the wild-type strains when comparing the number of clusters per sample across different doses, but due to the overall low number of clusters this link could only be reliably established in the wild-type (as it had the highest amount of samples tested). We have added a relevant Figure to Sup Figure S7.

4. “In C. elegans, ~6 % of all IR-induced mutations are clustered, an effect generally not observed in unchallenged C. elegans strains propagated over generations”: Can you make a direct comparison? What proportion of mutations are clustered in unchallenged strains?

Answer: As we reported in a publication studying mutation accumulation in DNA repair deficient strains (https://journals.plos.org/plosone/article?id=10.1371/journal.pone.0250291), we did not observe any clustering in wild-type lines even after propagating nematodes for 40 generations. Hence we did not expect any clustering in the untreated strains after just one generation. Also mutation numbers in control samples not treated with ionizing irradiation are too low to assess clustering. We have amended a relevant statement in the text and added clustering information across different genetic backgrounds to Supplementary Data (Supplementary Figure S7 (top left panel), Supplementary Table S2).

5. The link of the github repository refers to a different project.

Answer: Thank you, we have amended the link path.

6. Could you please share the VCF files with the readers? This will simplify reproducibility and potential future use of the data set in the community. If those files are already available in your previous publication, could you please make a simple list of how to find the VCF files relevant for this publication?

Answer: Following this comment, we have added a mutation counts table to the Supplementary File 1, and explicitly stated the link to the relevant VCF collection uploaded as part of our previous publication in the Supplementary Information section.

7. Would it be possible to include in the github repository everything needed to easily reproduce (at least the main) manuscript figures? (E.g., it seems the scripts expect various input files and parameterisation files, but I did not find those files in the repository.)

Answer: We have streamlined the codes appearing on the Github page and added more instructions on data preparation.The scripts in the Github repository expects the user to have downloaded the above mentioned VCF data as well as the supplementary table of this publication.

8. The methods should be ideally summarized in the methods/supplementary text of this paper in order to enable reproducibility without the need to look for many details and parameterization described in the previous paper. In particular, it would be good if the authors could describe in more detail how the variant filtering against a panel of unrelated samples was performed, as well as the cut-offs on coverage, read support of the variant in the test and control samples, and overlap with other genomic variants.

Answer: As mentioned in the Methods section, all the variant calling and filtering for this manuscript was performed in accordance with the procedure used in our previous publication (https://www.nature.com/articles/s41467-020-15912-7 ), Nevertheless, we have now added the relevant details to the Supplementary Information section of this manuscript (clearly referencing our previous paper). 

9. Could you please (in the discussion) briefly comment of similarities and differences of your results to those of mutational signatures of ionizing radiation observed in the following publications:

o (Li et al., 2020) https://www.nature.com/articles/s41467-019-14261-4

o (Kageyama et al., 2020) https://www.thegreenjournal.com/article/S0167-8140(20)30856-2/fulltext

o (Youk et al., 2021) https://www.biorxiv.org/content/10.1101/2021.01.12.426324v1.full

These new papers and preprints as well as a recent paper on thyroid carcinomas induced by the Chernobil disaster are now cited in the introduction by adding the underlined sentences below. Also in the discussion (text further below), we add a sentence highlighting that IR induced SV (in contrast to SV, and indels) could not be detected in some recent studies.

More recent studies, analysing thyroid carcinomas, linked to Iodine-131 exposure related to the Chernobyl disaster failed to detect increased SNV levels, indels and SVs being increased [19]. Human organoid exposure to IR similarly only led to increased indels and SVs [20]. Radiation induced tumours induced in p53 mutant mice SVs levels were reported to be rare, indels and SV being detectable [21] .

‘In summary, while these analyses provide insights into mutation rates and mutation types, their comparison is complicated by the types of analysis applied, differences in effects of chosen radiation sources, DNA repair status, tissue type, and possible organismal differences. Also, these studies do not provide insight into the relative contribution of various DNA repair pathways counteracting IR induced mutagenesis.’

added to the discussion:

‘The highest proportion of mutants being SVs is in line with the analysis of IR induced secondary tumours [17]. In contrast, increased SVs could not be detected in thyroid carcinomas, linked to the Chernobyl disaster [19] and irradiated human organoids [20]. ‘

Minor comments

1. Top of the page 10: the formula has some missing characters.

2. Page 13: incorrect font.

thanks, fixed

3. Page 14: a typo in “A Indeles”.

thanks, fixed

4. Page 14: “Finally the proportion of indels was increased in rev-1 and pol-h translesion synthesis defective strains”: Why proportion? (The figure seems to show mutation counts.)

thanks, we now say that ‘the number of indels’ was increased...

Reviewer #2: In this study, authors irradiate the germ cells of C.elegans in wild-type and DNA repair or damage response mutant animals and perform whole genome sequencing on the progeny which have undergone clonal expansion of mutations from zygotes. The study is methodologically sound and the analysis that was performed to describe the nature of these mutations generated by ionizing radiation (IR) is thorough. This is certainly an important and comprehensive study and the data is a valuable resource to the genomics community. While overall it is surprising that the radiation does not have a more substantial effect on mutagenesis in C.elegans, some interesting observations were made. The data suggests that global NER is important for repair of IR induced damage and that error prone bypass of IR lesions by translesion synthesis polymerases cause SNVs. A specific dinucleotide substitution pattern of mutations from irradiation was also identified. Minor comments are made below.

Minor

1. As the human genome is ~30x larger than the C.elegans genome, 3 replicates per parent might not result in enough mutations to make conclusions for lower frequency mutations such as SVs. This possibility should be added to the discussion.

Thanks for pointing this out, we now added two sentences (these are underlined below) acknowledging that our study is limited by us only analysing three replicates of each sample. The relevant section of the amended discussion is pasted below:

‘In line with expectations of IR induced mutation rate estimates in mouse and C. elegans germ cells [8,9,36] and from sequencing the progeny of irradiated mice, budding yeast and several plant species (for instance [10–16]), we confirmed that IR-induced mutagenesis is surprisingly low, with 80 Gy inducing an average of 36.6 SNVs, 1.3 DNVs, 4 indels and 1.4 (SD=0.13) SVs in the 100 million base C. elegans genome. We typically did experiments in triplicates, assessing samples without, and with an intermediate and high dose of IR. Clearly, increased statistical power would require the analysis of a larger number of samples. It is impossible to significantly increase the dose of IR as this would lead to 100% embryonic lethality. Broadly, 80 Gy leads to a 30 to 50% reduction of survival of irradiated progeny, with up to 10-fold lower survival in C. elegans DNA repair mutants [37–42].’

2. Since irradiation affects survival of progeny, can you discuss the possibility that the model system selects for progeny that accumulated less mutations? In the discussion on page 20, authors say this is not the case as apoptosis compromised ced-4 mutants do not have increased mutation burden. Can you preclude other possibilities such as necrosis or loss of capacity for self-fertilization?

Answer: We now added a sentence (underlined) and slightly changed our wording (inderlines) on pape 19, second last paragraph. We now state much clearer that we might underestimate mutation rates; us, in our experimental procedure, not sequencing gDNA from single embryos. 

‘On the other hand, while a large number of C. elegans fusion chromosomes are known to be viable in a heterozygous state and are being used as balancer chromosomes [48], aneuploidy of autosomes, typically caused by meiotic segregation defects, leads to embryonic lethality [49]. It is thus possible that we fail to detect a subset of some chromosomal fusions or the loss of large chromosomal fragments resulting from unrepaired DSBs, if these events lead to a near aneuploid state and consequently to embryonic or larval lethality. In other words, we might fail to detect highly mutagenised genomes because affected embryos die, and do not sire progeny needed for whole genome sequence analysis. Detection of mutational events leading to embryonic lethality would require single embryo or single zygote whole genome sequencing, studies we will pursue in the future.’

Figure 1C – Since COSMIC signature 3 and SBS40 are non-distinct signatures, the cosine similarity may appear similar to that of the humanized C.elegans irradiation signature without a genuine biological relationship between the signatures. You should compare the cosine similarity of simulated mutations to signature 3 and 40 and see if the value is lower.

Answer: We appreciate that flat mutational signatures without distinctive features would show high similarity to each other. We decided to draw value from the fact that the base substitution signature we extracted from IR-treated C. elegans lines also did not exhibit strong imbalances in mutation preference, making it similar to those flat signatures often observed in cancers. It is not clear to us what ‘simulated mutations’ refers to. We think that the reviewer meant comparing the IR signature we observe to a random mutation distribution. We note, as elaborated in our 2018 (Meier at al., ) paper, that trinucleotide prevalences between humans and C.elegans genomes vary, thus a random distribution of mutations would be rather different between the two species. 

4. Figure 2A – Add p-value and R-squared for line of best fit linear regression. It does not look like irradiation significantly induces indels and SNVs.

Answer: We have added the Pearson correlation coefficients and their corresponding p-value to Figure 2A.

5. Figure 2B – The figure legend describes values as fold change but the y-axis is number of mutations. Can you add ‘wild-type level’ on the dotted line (like in figures 5A and 5B) to make the comparison to wild-type clearer.

Thanks, we added this to Figure 2B and amended the legend.

6. Figure 5 – Can you perform simulations or bootstrapping analysis to see what proportion of mutations would fall in clusters by chance such as randomly shuffling mutations in trinucleotide preserved manner? Or simply, do non-irradiated animals also display mutation clusters?

As outlined in the response to the other reviewer (4) we previously reported in a publication studying mutation accumulation in DNA repair deficient strains (https://journals.plos.org/plosone/article?id=10.1371/journal.pone.0250291) that we did not observe any clustering in the wild-type lines even after propagating lines for 40 generations. Hence we did not expect any clustering in the untreated strains after just one generation. We have amended a relevant statement in the text and added clustering information across different genetic backgrounds to Supplementary Data.

7. Figure 5 – Where in the genome do mutation clusters lie? For example, are they associated with heterochromatin or other epigenetic marks? Were there any genomic sites of mutation clustering that occurred in more than 1 sample tested.

The number of mutational clusters observed in the experiments was too low to try to make an inference about the genomic location of the clusters. We did not observe recurrent mutation clusters (we have added another Supplementary Data object describing mutations in clusters across different genomic backgrounds which demonstrates this (also see response 4 to previous reviewer)).

B) Reasons for our work not constituting dual publication: 

We note that our current manuscript uses the same primary sequence source data we deposited as part of our Nat Commun. 2020 May 1;11(1):2169. doi: 10.1038/s41467-020-15912-7. We clearly indicate this in the abstract, the results section, and the discussion. In the Nature Communication paper, we provided a high-level overview (and comparison) of the mutagenic consequences of exposure to 11 genotoxic agents, one agent being ionizing radiation (IR).

Given the immense importance of understanding IR damage, we here provide a much more detailed and comprehensive analysis of the mutagenic consequences of IR as part of this manuscript. Such detailed, in-depth analysis of IR-induced mutagenesis was not possible in our Nat Communication paper and is a valuable resource for researchers studying IR response and mutagenesis. Our five main figures, the three supplementary files and the 8 supplementary figures are all new and are all based on new analysis. None of the figures of this manuscript was shown in our Nature Communications publication. Introductory Figure 1 provides a refined analysis of the IR-induced mutational signature in DNA repair proficient C. elegans and a comparison to radiation-associated signatures observed in human cancer. Importantly, absolute mutation numbers and the variation between IR exposure experiments are clearly visualized. Figure 2 provides a refined analysis of DNA repair mutants that showed increased numbers of single nucleotide variants (SNVs), multi nucleotide variants (MNVs), indels and structural variants (SVs) compared to wild-type. The analysis is new and (in contrast to our Nature Communications paper) only considers IR exposed samples. Figure 3 provides a refined analysis of the extent of change in mutational signatures in various DNA repair mutants. Importantly, Figure 4 provides a new, comprehensive analysis of dinucleotide variants induced by radiation, compared to those induced by UV and cisplatin exposure. Figure 5, summarizes for the first time the extent of mutational clustering upon IR exposure, which is surprisingly low and not overtly affected by the deficiency of various DNA repair pathways analysed in our studies. Suppl Files 1-3 show a comprehensive analysis of the genomic location of mutations, mutational profiles and dose response in wild-type and DNA repair mutants. Suppl Fig S1 compares indel frequencies in our experimental system and IR induced cancers, Suppl Fig S2-6 provide a comprehensive overview of absolute numbers and types of mutation resulting from IR exposure, stratified by DNA repair pathway. Suppl Fig S7 and 8 provide supporting data of our analysis on mutational clustering (Figure 5).

Importantly, in this manuscript we also discuss our data in detail, providing a comprehensive review of previous literature and also providing a detailed and comprehensive discussion of our data and its relevance to the field. The in-depth description of IR induced mutagenesis will provide a basis for a more targeted design of analogous studies in mammalian cells.

Finally, we point out that both reviewers strongly supported our work, and both second that our manuscript is novel and distinct from our previous publication.

---

## [Editor Report · Decision Letter 1]

23 Sep 2021

C. elegans genome-wide analysis reveals DNA repair pathways that act cooperatively to preserve genome integrity upon ionizing radiation

PONE-D-21-18200R1

Dear Dr. Gartner,

We’re pleased to inform you that your manuscript has been judged scientifically suitable for publication and will be formally accepted for publication once it meets all outstanding technical requirements.

Kind regards,

Sue Cotterill

Academic Editor

PLOS ONE
---

## [Editor Report · Acceptance letter]

28 Sep 2021

PONE-D-21-18200R1 

*C. elegans* genome-wide analysis reveals DNA repair pathways that act cooperatively to preserve genome integrity upon ionizing radiation 

Dear Dr. Gartner:

I'm pleased to inform you that your manuscript has been deemed suitable for publication in PLOS ONE. Congratulations! Your manuscript is now with our production department. 

Kind regards, 

on behalf of

Dr Sue Cotterill 

Academic Editor

PLOS ONE